# Long-Term, Health-Related Quality of Life after Open and Robot-Assisted Ivor-Lewis Procedures—A Propensity Score-Matched Study

**DOI:** 10.3390/jcm9113513

**Published:** 2020-10-30

**Authors:** Anne-Sophie Mehdorn, Thorben Möller, Frederike Franke, Florian Richter, Jan-Niclas Kersebaum, Thomas Becker, Jan-Hendrik Egberts

**Affiliations:** Department of General, Abdominal, Thoracic, Transplantation and Pediatric Surgery, University Hospital Schleswig-Holstein, Campus Kiel, Arnold-Heller-Straße 3, 24105 Kiel, Germany; anne-sophie.mehdorn@uksh.de (A.-S.M.); thorben.moeller@uksh.de (T.M.); Frederike.Franke@uksh.de (F.F.); florian.richter@uksh.de (F.R.); jan-niclas.kersebaum@uksh.de (J.-N.K.); thomas.becker@uksh.de (T.B.)

**Keywords:** esophagectomy, esophageal cancer, Ivor-Lewis procedure, robotic surgery, health-related quality of life

## Abstract

Esophagectomies are among the most invasive surgical procedures that highly influence health-related quality of life (HRQoL). Recent improvements have helped to achieve longer survival. Therefore, long-term postoperative HRQoL needs to be emphasized in addition to classic criterions like morbidity and mortality. We aimed to compare short and long-term HRQoL after open transthoracic esophagectomies (OTEs) and robotic-assisted minimally invasive esophagectomies (RAMIEs) in patients suffering from esophageal adenocarcinoma. Prospectively collected HRQoL-data (from the European Organization for Research and Treatment of Cancer Core Quality of Life Questionnaire-C30 (EORTC QLQ-C30)) were correlated with clinical courses. Only patients suffering from minor postoperative complications (Clavien–Dindo Classification of < 2) after R0 Ivor-Lewis-procedures were included. Age, sex, body mass index (BMI), American Society of Anesthesiologists physical status-score (ASA-score), tumor stage, and perioperative therapy were used for propensity score matching (PSM). Twelve RAMIE and 29 OTE patients met the inclusion criteria. RAMIE patients reported significantly better emotional and social function while suffering from significantly less pain and less physical impairment four months after surgery. The long-term follow up confirmed the results. Long-term postoperative HRQoL and self-perception partly exceeded the levels of the healthy reference population. Minor operative trauma by robotic approaches resulted in significantly reduced physical impairments while improving HRQoL and self-perception, especially in the long-term. However, further long-term results are warranted to confirm this positive trend.

## 1. Introduction

Esophageal cancer (EC) is, with increasing incidence, becoming among the most common malignancies worldwide [1,2]. The therapy of choice for locally-resectable EC still is surgery [3,4]. If indicated, additional perioperative radio chemotherapy is applied [3,4]. Depending on the localization of the tumor, surgery is performed as a two- or three-cavity approach with an intrathoracic (Ivor-Lewis) or cervical (McKeown) anastomosis, respectively [5,6,7,8]. Since the procedure involves at least two cavities, the surgical approach is important. Perioperative surgical and oncological results, as well as the physical and psychological burden of surgery, highly influence the long-term, postoperative, health-related quality of life (HRQoL), postoperative course, and outcomes [5,9,10,11,12,13].

Since the 1940s, open transthoracic esophagectomies (OTEs) have been the gold standard for EC, achieving solid oncological results but also causing high postoperative complication rates, prolonged postoperative courses, major physical impairments, and long-lasting physical deterioration for up to ten years after surgery [3,6,11,12,13,14]. In the 1990s, minimally-invasive esophagectomies (MIEs) and so-called hybrid-MIEs (HMIEs) were introduced to overcome the shortcomings of OTE [4,13]. While achieving fewer postoperative complications and comparable oncological results, MIE and HMIE are associated with risks, complications, and limitations of their own [3,4,5,15,16,17,18,19]. For the first time in 2003, Kerstine described a robot-assisted MIE (RAMIE) [20], which appeared to overcome the limitations of not only OTE but also MIE and HMIE [3,7,20]. The advantages of RAMIE range from less intraoperative trauma, shorter hospital stays, and cosmetically more-appealing surgical incisions to more comfortable positions for the operating surgeon, a better visualization of the operative field, and greater degrees of freedom for the instruments during surgery, especially in narrow spaces like the mediastinum [7,17,21,22]. In specialized surgical centers with robotic systems and consecutive experience, RAMIE has become a standard surgical procedure for patients with EC [4,7,22,23]. Postoperative courses and oncological radicalness after RAMIE have been shown to be comparable to OTE [3,4,10,15,16].

However, no clear definition of the surgical gold standard for EC exists, and different techniques are still applied worldwide [5,24,25,26,27,28]. Therefore, OTE is often used to as reference procedure, even in large trials like the TIME-, MIRO- and ROBOT-trials [4,7,13,26,27,29,30]. Furthermore, terms for esophagectomies, e.g., MIE, HMIE, RAMIE, Ivor-Lewis, and McKeown, are used variably in the literature [11,25,31,32]. Additionally, no surgical approach has unambiguously been proven to be superior compared to the others [33,34]. 

However, EC patients not only suffer from the diagnosis of cancer but must also cope with the insecurity, physical changes, and impairments associated with the disease. Severe fear of surgery, physical impairments, and the reduction of HRQoL after unavoidable surgery must be managed. EC patients even choose reduced survival time over postoperative complications, accepting alternative, potentially non-curative treatments with fewer complications to avoid surgery [35]. Such therapies, including watchful waiting, radio chemotherapy, and brachytherapy, are physically and psychologically challenging, reduce HRQoL for up to five years after application, and are associated with risks of tumor progression, perforation, bleeding, and post-interventional fistula, among others [36,37,38,39,40,41].

Though RAMIE has led to promising short-term postoperative results [3,7,31], little is known about its long-term influence on physical functions, oncological outcomes, and, especially, HRQoL. Long-term improvements in HRQoL after esophagectomy are gaining importance alongside traditional outcome measures such as improved oncological results, the Clavien–Dindo Classification, and long-term survival. HRQoL is increasingly being used to evaluate new surgical techniques [3,7,14,17,21,22,42,43,44]. However, to the best of our knowledge, no study has analyzed and evaluated long-term HRQoL after RAMIE. Therefore, we aimed to compare postoperative courses in EC patients undergoing OTE and RAMIE with curative intention in a German tertiary referral center, with a special focus on the short- and long-term effects on HRQoL. As the presence of complications can negatively or positively influence the subjective perception of the postoperative course, only patients without major postoperative complications (Clavien–Dindo Classification of < 2) were included and compared in a propensity score matching (PSM) analysis for up to 18 months after surgery [44]. 

## 2. Material and Methods

### 2.1. Data Collection

A database prospectively collecting the HRQoL-data of all patients undergoing surgery is maintained at the Department of General, Visceral, Transplant, Thoracic, and Pediatric Surgery, University Hospital Schleswig-Holstein, Campus Kiel, Germany. In brief, the European Organization for Research and Treatment of Cancer Core Quality of Life Questionnaire-C30 (EORTC QLQ-C30) and a tumor-specific module that cover different aspects of physical function and psychological status are used [21]. Patients receive the questionnaire prior to surgery and four and 18 months postoperatively. Questionnaires are returned in pre-stamped envelopes, at no cost to the patients. The EORTC QLQ-C30 is an internationally-validated questionnaire covering different functional aspects of self-perception (such as physical, cognitive, emotional, role, and social function) and QoL (including global health status, postoperative function, and impairments such as pain, insomnia, appetite loss, vomiting, constipation, diarrhea, and financial difficulties). Items are classified in four (“not at all” to “very much”) and seven categories (“very poor” to “excellent”), respectively. High scores in the functional aspects represent high levels of functioning (i.e., good QoL), whereas low scores in the impairment categories represent few side effects [45]. Results are compared to a healthy reference population [46]. The data of the healthy reference population are provided by the European Organization for Research and Treatment of Cancer [46]. 

Demographic data and clinical courses of EC patients were retrospectively retrieved from hospital’s in-house patient files. All patients gave written informed consent for inclusion in this study and the use of their data. The local ethics committee provided written approval (AZD421/13, D451/19). The study adhered to the principles of the Declarations of Helsinki and Istanbul. Only de-identified data were used for further analysis. Patient data included age, gender, body mass index (BMI), American Society of Anesthesiologists physical status-score (ASA score), and comorbidities such as coronary heart disease, heart insufficiency, myocardial infarction, arterial hypertension, chronic obstructive pulmonary disease (COPD), diabetes, history of smoking, and alcohol consumption. These comorbidities have been shown to be associated with an increased risk of EC.

### 2.2. Inclusion Criteria

Patients with adenocarcinoma of the esophagus who underwent Ivor-Lewis OTE (2005–2010) or RAMIE (2013–2017) at the Department of General, Visceral, Thoracic, Transplantation, and Pediatric Surgery, University Hospital Schleswig-Holstein, Campus Kiel, Germany, were included. A complete HRQoL follow-up was mandatory to achieve a valid statement on postoperative courses. Exclusion criteria were age <18 years, any other procedure than Ivor-Lewis esophagectomy, the conversion of robotic procedures, any type of carcinoma other than adenocarcinoma, complications Clavien–Dindo Classification ≥ II, and R1- or R2-resections. 

### 2.3. Surgical Procedures

Because OTE has long been the surgical gold standard for EC, it was used as the reference procedure against which RAIME patients were compared. We aimed to achieve homogenous and comparable study populations and to exclude the influence of the learning curve on a new technique. 

The OTE was performed as previously described [3,6,23]. In brief, a median incision was made, followed by the transhiatal mobilization of the stomach and D2-lymphadenectomy before the formation of a gastric tube. If necessary, omentectomy or further lymphadenectomy were performed. After the completion of the abdominal part, the patient’s position was changed to the prone position, and a right thoracotomy between the 5th and 6th rib was achieved while performing one-lung ventilation. The mobilization and resection of the esophagus were completed with an en-bloc lymphadenectomy, and a hand-sewn or stapler-based anastomosis was performed [3,6].

For the abdominal and thoracic approaches in RAMIE, the da Vinci Si^®^ (2013–2014) or da Vinci Xi® (2014–2017) systems were used [22,23]. Patients were initially put in a supine or French position and later changed to swimmer’s position for the thoracic part. For the abdominal part, four ports were placed; after liver retraction and mobilization, D2-lymphadenectomy was performed before the mobilization of the stomach and the opening of the hiatus to release the esophagus prior to construction of the gastric tube. Since 2017, feeding tubes have been implanted in the first jejunal loop to secure enteral nutrition. For the thoracic approach, four trocars were positioned using a left thoracic approach. The esophagus was dissected from the surrounding tissue at the level of the hiatus down to the pericardium and cranially at the level of the azygos vein using a linear stapler. The gastric tube was then carefully pulled up, and a stapler-based anastomosis was performed transorally [20,22,23,47]. 

Postoperatively, all patients were transferred to the intensive care unit (ICU) and extubated as soon as possible. A histopathological analysis of the resected specimens was performed by a board of specialized pathologists. Pre- and post-operative presentation in an interdisciplinary tumor board was mandatory, and adjuvant (radio)chemotherapy was given if recommended. Postoperative courses were documented in the in-house patient files. Postoperative complications were classified according to Clavien–Dindo Classification [48]. The time of surgery, type of anastomosis, use and type of stapler, size of tumor and lymph node harvested, and (neo)adjuvant (radio)chemotherapy were assessed. The Union for International Cancer Control (UICC) TNM-staging version 8 was used to classify tumor stage (UICC-guidelines for EC, version 8).

### 2.4. Outcome Measures

The primary endpoint was the completion of OTE or RAMIE and a postoperative course without complications (Clavien–Dindo Classification ≥ II). Secondary endpoints included the overall HRQoL for up to 24 months postoperatively and overall and disease-free survival.

### 2.5. Statistical Analysis

Qualitative data are presented as means ± standard deviations (SD) and ranges, evaluated using the Chi-square test. Quantitative data are presented as percentages, evaluated using Student’s t-test. Survival data were analyzed and interpreted using the Kaplan–Meier method and log-rank tests [49]. Survival was defined from surgery to last contact or death, whichever occurred first. HRQoL data were pooled for 4 months (3 and 6 months) and 18 months (12 and 24 months) postoperatively and analyzed according to the EORTC-scoring manual, as previously described [21]. Student’s *t*-test was used to compare the cohorts. *р*-values < 0.05 were considered significant. Age, sex, BMI, TNM-stage, ASA-classification, and (neo)adjuvant radio- and chemotherapy were used as matching criteria to perform the PSM analysis. The propensity matching score was 0.1. All statistical analyses were performed using IBM® SPSS Statistics Version 25 for Windows (IBM, Somers, NY, USA). Graph Pad Prism was used to present data. 

## 3. Results

Between 2005 and 2017, 29 OTE patients and 12 RAMIE patients met the inclusion criteria and were included in the final analysis (Figure 1). Cohorts were comparable regarding demographic data, with no significant differences (Table 1). Most patients were male (83.3%and 86.2%, respectively) and overweight (BMI 26.5 ± 4.6 and 28.3 ± 4.5 kg/m^2^, respectively). The average age was 64.5 ± 9.1 vs. 61.5 ± 8.2 years, respectively). Though the differences were not significant, RAMIE patients smoked more often than OTE patients (41.7% vs. 24.1%, respectively), suffered more often from COPD (16.7% vs. 10.3%, respectively), and had been classified as sicker according to the ASA-classification. OTE patients suffered more often from arterial hypertension than RAMIE patients (55.2% vs. 25.0%, respectively) and diabetes mellitus (17.2% vs. 0%, respectively). Furthermore, RAMIE patients more often received both neoadjuvant chemo- and radiotherapy and adjuvant chemotherapy. 

Surgical times, postoperative complications, UICC-stage of tumors resected, and length of stay at the hospital were comparable (Table 2). Interestingly, tumors resected by RAMIE tended to be larger (diameter of 31.9 ± 11.7 mm for RAMIE patients vs. 20.6 ± 20.9 mm for OTE patients). In addition, RAMIE achieved a significantly higher lymph node yield (31.0 ± 10.0 for RAMIE patients vs. 18.7 ± 12.1 for OTE patients; *p* = 0.004). Overall and disease-free survival were comparable between the RAMIE and OTE cohorts (*p* = 0.279 and *p* = 0.510, respectively) (Figure 2). The follow-up for RAMIE patients was shorter due to the more recent inclusion period. 

HRQoL was assessed four and 18 months after surgery and compared to a healthy reference population. As this was an inclusion criterion, follow-up was 100%. The results for the different items used to describe QoL, HRQoL, and self-perception and self-esteem in the EORT QLQ-C30 can be seen in Figure 3A–E and Figure 4A–J. As expected, surgery influenced QoL and symptoms in both cohorts, and patients perceived symptoms to be stronger than the general population (Figure 4A–J). However, four months after surgery, the overall QoL was better after RAMIE than after OTE (Figure 4A). Additionally, RAMIE patients reported less fatigue, nausea, vomiting, pain, dyspnea, appetite loss, and diarrhea in the early postoperative follow-up (Figure 4B–I). In the long-term, RAMIE patients seemed to recover better than OTE patients, reporting lower levels of impairment and deterioration up to 18 months after surgery. Interestingly, RAMIE patients suffered from fatigue significantly less often in the long-term (Figure 4B). Neither cohort reported a change in financial difficulties after esophagectomy in the short- or long-term (Figure 4J). It is noteworthy that RAMIE patients reported QoL levels that were similar to the healthy reference population in the long-term, while the QoL in OTE patients was still reduced 18 months after surgery; only a small improvement in QoL was reported during the postoperative course. Other body functions, such as dyspnea, diarrhea, fatigue, nausea/vomiting, and postoperative pain, were almost at the level of the healthy reference population after RAMIE. 

In the early postoperative follow-up, patients in both cohorts felt impaired regarding all physical functions, reporting levels beneath those of the reference population (Figure 3A–E). Nonetheless, emotional and social function were significantly better after RAMIE, with emotional function almost reaching the level of the reference population (Figure 3C,E). Furthermore, RAMIE patients tended towards higher physical, role, and cognitive functions (Figure 3A,B,D). At the long-term, 18-month follow-up, RAMIE patients reported highly-improved physical, role, emotional, and social functions, even feeling recovered to the level of the reference population (Figure 3A–E). In contrast, OTE patients only showed moderate improvements during the same follow-up period. In particular, role function was highly and significantly better after RAMIE, with additional significant differences regarding social and emotional function in favor of RAMIE. 

A PSM analysis was performed to achieve a more precise comparison of the influence of the surgical approach on HRQoL. Out of 29 OTE and 12 RAMIE patients, 22 patients, eleven from each cohort, were included according to the matching criteria chosen. The PSM cohorts were comparable regarding demographic data, with few significant differences between RAMIE and OTE patients (Table 3). Most patients were male (81.8% vs. 72.7%, respectively) and of similar age (64.4 ± 9.5 vs. 63.2 ± 6.0 years, respectively). RAMIE patients smoked more often than OTE patients, whereas OTE patients reported a tendency towards more arterial hypertension. Furthermore, RAMIE patients received more adjuvant chemotherapy. It is noteworthy that all anastomoses in OTE were stapler-based, whereas 54.5% of the anastomoses in RAMIE were hand-sewn. Again, tumors tended to be larger in RAMIE patients, and their lymph node yield was significantly higher compared to OTE patients (29.9 ± 9.8 vs. 18.1 ± 13.8, respectively; *p* = 0.031). The length of surgery and hospital stay were comparable (Table 4).

Regarding HRQoL, the PSM revealed more obvious differences between RAMIE and OTE patients. Early in the postoperative course, RAMIE patients reported significantly better QoL and significantly less pain (Figure 5A,D). Additionally, RAMIE patients showed a tendency towards less physical impairment regarding other symptoms (Figure 5B–I), with an improvement in QoL and the amelioration of symptoms over time (Figure 5A–I). Eighteen months after surgery, the QoL of RAMIE patients improved further and was comparable to that of the general population (Figure 5A). At this time, appetite loss was significantly reduced in RAMIE patients compared to OTE patients (Figure 5G). Interestingly, there was little change in postoperative pain levels in the long-term among RAMIE patients, potentially already reflecting low postoperative pain levels after RAMIE and improved compared to OTE patients. The type of procedure, however, did not influence the financial situation in the long-term in either cohort. 

Four months after surgery, both cohorts reported deterioration regarding body functions (Figure 6A–F). However, RAMIE patients reported less impairment, with function levels closer to those of the general population. In addition, social and emotional functions were significantly better among RAMIE patients (Figure 6C,F). In both cohorts, functions improved over time. However, long-term improvements were much stronger among RAMIE patients, with physical, role, and social functions reaching 90% (90.0 ± 8.6%, 90.0 ± 14.9%, and 90.0 ± 9.3%, respectively; Figure 6A,B,E). These body and emotional functions overtook function levels reported by the general population and were significantly better than those reported by OTE patients (Figure 6A–C,E). 

## 4. Discussion

The diagnosis of EC is still troubling, and esophagectomies are among the most challenging surgical procedures, potentially causing many physical changes and impairments [3,14,42]. It was hoped that the introduction of RAMIE would lead to improved postoperative courses compared to conventional and MIE procedures [7,20]. From a surgical point of view, RAMIE offers major additional technical advantages in the narrow mediastinum that can enable, for example, high lymph node yields, the more comfortable and technically easier creation of intrathoracic anastomoses using smaller thoracic incisions, and less severe post-thoracotomy syndromes [3,7,16,17,18,20,22,23]. Using tissue- and nerve-sparing surgery, functional impairments and, thus, impairments of body function are assumed to result in fewer long-lasting impairments and thereby improved long-term body function after RAMIE [11].

Our study aimed to compare the postoperative HRQoL after traditional OTE and RAMIE Ivor-Lewis procedures. To obtain an homogenous cohort of patients and eliminate the influence of comorbidities and perioperative complications, we chose to only include patients who suffered from adenocarcinoma of the esophagus, had experienced only minor complications after surgery, and who presented with complete postoperative HRQoL data for up to 18 months after surgery. To further exclude other biases, included patients were matched for age, sex, BMI, tumor stage, and application of perioperative radio chemotherapy. This analysis further focused on short- and long-term postoperative functions and different body functions after OTE and RAMIE. Since OTE is a well-established procedure and the introduction of novel techniques and procedures always has to deal with a learning curve, we only included full OTE procedures and, for better comparison, full RAMIE procedures in our analysis [25,26,29,32]. Since complications with a Clavien–Dindo Classification > II are often used as a stopping point, we chose to use it as one of our inclusion criteria [7,30]. 

Our analysis in patients with minor postoperative complications clearly identified better HRQoL four months after RAMIE compared to OTE. The 1:1 PSM analysis further emphasized this result and revealed significantly better HRQoL and reduced postoperative pain after RAMIE in the short-term follow-up. We consider this to be the result of the less harmful surgical approach, the comparably few and small postoperative reductions in the different body functions, and reportedly lower postoperative pain. Early after surgery, RAMIE patients even reported an HRQoL that was comparable to the healthy reference population, thus not considering surgery the major issue that it has traditionally been. The ROBOT study was the first to compare postoperative HRQoL and perioperative pain after OTE and RAMIE procedures [7], and it found a higher postoperative HRQoL, better body functions, and greater functional recovery immediately and six weeks after hybrid-RAMIE. However, the procedures included in the ROBOT trial consisted of hybrid procedures combining laparoscopic abdominal and robot-assisted thoracic approaches. The ROBOT study was therefore only partly suitable for a comparison of HRQoL after full RAMIE, as performed in our analysis. Recently, Sakaria et al. compared early postoperative HRQoL after full RAMIE and OTEand reported better HRQoL with RAMIE at four months after surgery [31]. However, they included thoracoabdominal, Ivor-Lewis, and McKeown RAMIEs, and they only reported results for a four-month follow-up, during which 20% of patients were lost [31]. To the best of our knowledge, no further study has analyzed or compared HRQoL after RAMIE versus OTE. Since terms, e.g., MIE, HMIE, and RAMIE, are often mixed and heterogeneous cohorts, including Ivor-Lewis, McKeown, Sweet, and other types of surgical approaches are often mixed, we consider our approach, which compared full RAMIE to the established gold standard OTE while only including Ivor-Lewis-procedures, appropriate. Furthermore, most studies have reported on hybrid procedures and avoided the full laparoscopic approach due to the limitations of laparoscopic surgery [28]. Our RAMIE cohort, on the other hand, underwent a full robotic approach after having completed the learning curve in hybrid RAMIE [22].

Early postoperative HRQoL is mainly influenced by the postoperative impairments of body functions, such as swallowing problems, a loss of appetite, and vomiting, as well as postoperative pain. After OTE, but also after HMIE using open thoracic approaches, strong and persistent thoracic pain—the so-called post-thoracotomy syndrome—significantly influences postoperative QoL [50,51]. Postoperative pain is mainly caused by the open approach, when rips and damage to intercostal nerves spreads [51]. In our analysis, which included a robotic thoracic approach, RAMIE patients reported significantly less pain compared to OTE patients in the PSM analysis. Both the MIRO and ROBOT trials reported reduced postoperative pain after MIE or robotic thoracic approaches, clearly attributing the main pain to the transthoracic approach [4,7]. Reduced postoperative pain has also been reported for up to one year after MIE, and this has been attributed to the significantly smaller and less invasive surgical approaches, reduced intraoperative trauma, and, thus, a lower incidence of post-thoracotomy syndromes [15,16]. In line with this, Sakaria et al. also reported a reduction in postoperative pain for up to four months after full RAMIE [31]. A surrogate for reduced postoperative thoracic pain may be dyspnea. Patients unambiguously reported less dyspnea after RAMIE in our analysis, which may suggest less postoperative thoracic pain. 

While body functions are best used to evaluate short-term results, self-esteem and self-perception are the best surrogates for long-term recovery after esophagectomy [52]. Postoperative HRQoL, self-esteem, and self-perception have gained importance in recent years and have become other factors when evaluating a procedure, especially when considering improved oncological results and longer survival rates [7,21,44]. Even early after surgery in our study, RAMIE patients reported superior postoperative HRQoL but also significantly better emotional and social functions. We assume that the latter were attributable to reduced intraoperative trauma, fewer postoperative impairments, and, therefore, better body functions and superior HRQoL. The early, positive postoperative trend of improved HRQoL after RAMIE persisted over time and was again reported 18 months after surgery. RAMIE patients in both the whole cohort and in the PSM analysis clearly stated a higher postoperative HRQoL in the long-term follow-up. Furthermore, an unambiguous improvement in physical, role, emotional, and social functions was detectable among RAMIE patients. In particular, role, emotional and social functions were significantly better compared to OTE patients and even overtook levels reported by the healthy reference population. The PSM analysis underlines the results of the whole cohort. 

As RAMIE is a relatively new surgical procedure, no long-term HRQoL follow-up data have yet been reported, including the ROBOT study and that by Sakaria et al. Therefore, our analysis is the first to report the impact of the robotic procedure on HRQoL. However, severe and long-lasting impairments regarding HRQoL, physical function, self-perception, and self-esteem have been reported for up to 10 years after OTE [14,42,53]. Mariette et al. further reported reduced social function for up to two years after HMIE [4]. In contrast, Taioli et al. identified superior social and emotional function in the long-term follow-up after MIE compared to OTE [15], while Mantoan et al. reported improved role function after HMIE compared to the healthy reference population [11]. A full recovery of HRQoL and physical and emotional function was reported by Qi et al. two years after an MI Ivor-Lewis procedure for squamous cell carcinoma [54]. Interestingly, they even demonstrated an “over recovery” of emotional and social function after surgery, with both functions being reported as superior to preoperative levels [54]. Klevebro et al., on the other hand, reported no differences in HRQoL when comparing OTE to MIE and HMI for up to one year after surgery [55]. These studies only allowed for a partial comparison to our results due to different surgical approaches, tumor entities, and study designs. Despite these limitations, they are the only existing potential comparisons. In addition to the improvements in self-perception, physical function and postoperative impairments were found to improve in the long-term follow-up in our RAMIE patients, who reported less pain and physical impairments such as vomiting, nausea, dyspnea, diarrhea, and loss of appetite or sleep compared to OTE patients. Fatigue, which highly influences postoperative courses, was reported significantly less often after RAMIE in the long-term. This may also be attributable to the improved postoperative HRQoL and self-perception. Additionally, not only surgery and the surgical approach chosen influence postoperative HRQoL, because complications also influence the latter. Heits et al. reported on the so-called “response shift” after prolonged postoperative courses due to major complications, while Scarpa et al. mentioned impaired postoperative HRQoL on the long-term in patients suffering from postoperative complications [50,56]. Most studies have only reported on major complications or have not differentiated between minor and major complications [34,56,57]. Rutegard et al. also reported on impaired HRQoL after major surgical complications in patients undergoing OTE [58]. Kaupilla et al., on the other hand, reported on the influence of medical complications for up to 10 years after surgery and surgical complications for up to five years after surgery [59]. By our inclusion criteria, we aimed to exclude this vagueness and the response shift influencing the results. Our inclusion criteria voluntarily only included patients suffering from minor complications in order to avoid response shift, an extensive influence of major complications following extensive or long therapy, e.g., esosponge-therapy, stent, and reoperation, on HRQoL [28,34,57,59]. However, in this context, it would be especially interesting to see if major complications differently influence HRQoL in RAMIE patients compared to OTE patients, especially in the long-term.

In our study, RAMIE was superior to the established OTE regarding body and physical functions in the short- and long-term follow-up. Due to the smaller surgical approaches, less-obvious scars, and fewer visible bodily changes during RAMIE, there is less of an impact on physical function, self-perception, and role function, and there is therefore a smaller influence on social and emotional function, which results in improved self-esteem [4]. 

Postoperatively, all oncological patients at our department are offered both tailored psycho-oncological therapy and rehabilitation after surgery. In addition to the surgical changes, this may further influence postoperative courses, HRQoL, self-perception, and self-esteem, especially in the long-term. Considering the understanding of Wikmann et al. regarding worse clinical courses after esophagectomy in patients with new-onset depression [10], we place special emphasis on postoperative HRQoL and self-perception after esophagectomy, and we consider our results to be encouraging. Our patients unambiguously reported better postoperative HRQoL and role function, as well as fewer impairments with regard to body functions after RAMIE, especially in the long-term follow-up. 

However, there were several limitations to our study. Due to the small number of patients fulfilling the inclusion criteria and the partly retrospective character of the study, the statistical power was limited. On the other hand, inclusion criteria were made quite strict to ensure two homogenous cohorts and thus achieve a more valuable comparison. Small demographic differences should have had only a minor, if any, influence on the results. Secondly, not all patients had been treated neoadjuvantly. Since we used neoadjuvant therapy as one of the matching criteria in the PSM, we excluded the influence of neoadjuvant radio chemotherapy on symptoms and function. Due to the relative novelty of the procedure, there were no valid long-term follow-ups (>18 months) regarding HRQoL or oncological outcomes after RAMIE. We also could not provide preoperative data on HRQoL. However, it is known that the diagnosis of a malignant tumor, the associated stress and fear, and neoadjuvant treatment negatively influence physical and psychological well-being [11,14,44]. We therefore do not consider preoperative HRQoL data as representative and feel that comparing patients to a healthy reference population is more useful. It is also well-known that postoperative complications influence postoperative courses, especially the perception of the postoperative course and HRQoL [4,21,50]. Patients learn to live with and adapt to problems and limitations. After a certain period of time, impairments are no longer perceived as such and may even be considered positively [36,43,50]. To exclude this bias, we only included patients with a Clavien–Dindo Classification of < II and are well-aware that our cohorts do not represent a normal cohort after esophagectomy. 

Patients who underwent robot-assisted esophagectomy reported fewer physical impairments and better HRQoL and body functions in short- and long-term (18 months) follow-ups compared to open esophagectomy patients, and they did so without experiencing a reduction in oncological outcomes. We therefore consider RAMIE to be a safe and, regarding its positive impact on long-term HRQoL and self-perception, preferable procedure for patients with adenocarcinoma of the esophagus. However, further long-term analyses are needed to verify this positive trend.

## Figures and Tables

**Figure 1 jcm-09-03513-f001:**
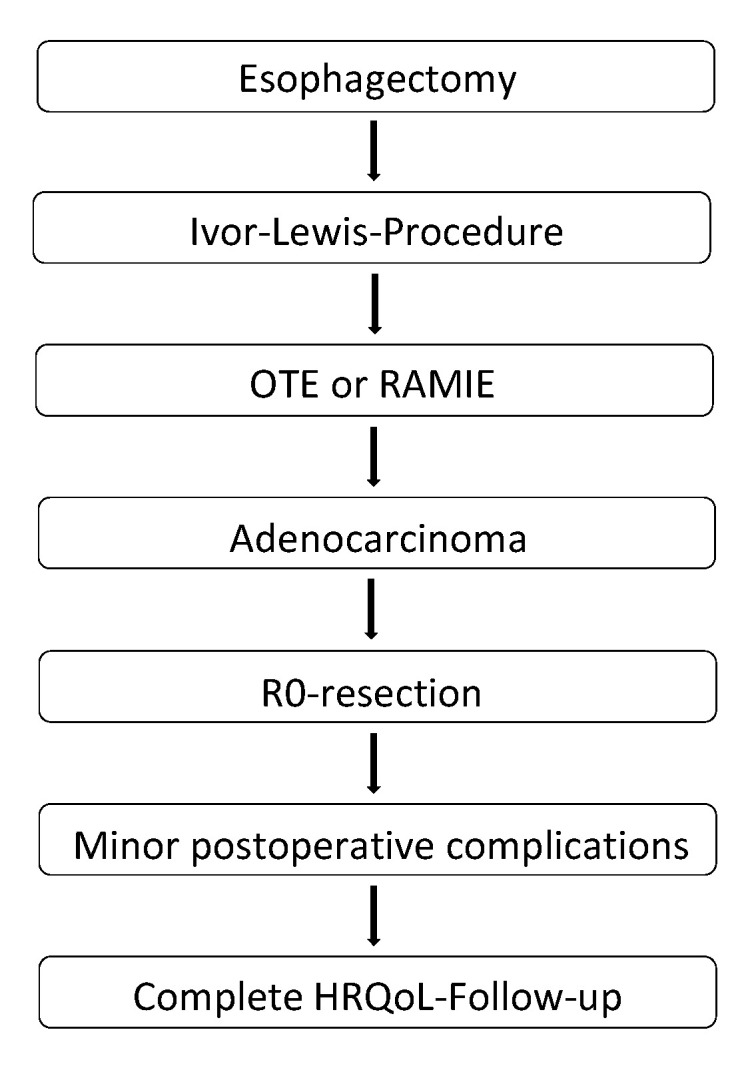
Flow chart of patient inclusion into the study. OTE: open transthoracic esophagectomy; RAMIE: robot-assisted minimally-invasive esophagectomy; HRQoL: health-related quality of life.

**Figure 2 jcm-09-03513-f002:**
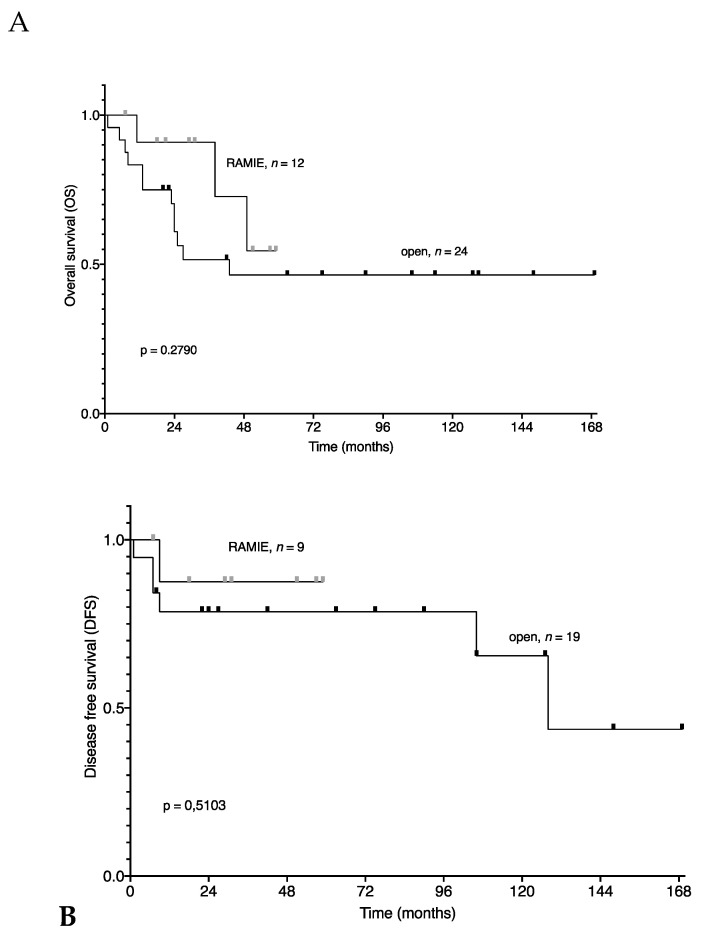
Survival stratified by the cohorts—RAMIE (red) vs. OTE (green). (**A**) overall survival; (**B**) disease-free survival. Kaplan–Meier survival curves and the log-rank test were used to compare survival. DFS: disease-free survival; OS: overall survival; OTE: open transthoracic esophagectomy; RAMIE: robot-assisted minimally-invasive esophagectomy.

**Figure 3 jcm-09-03513-f003:**
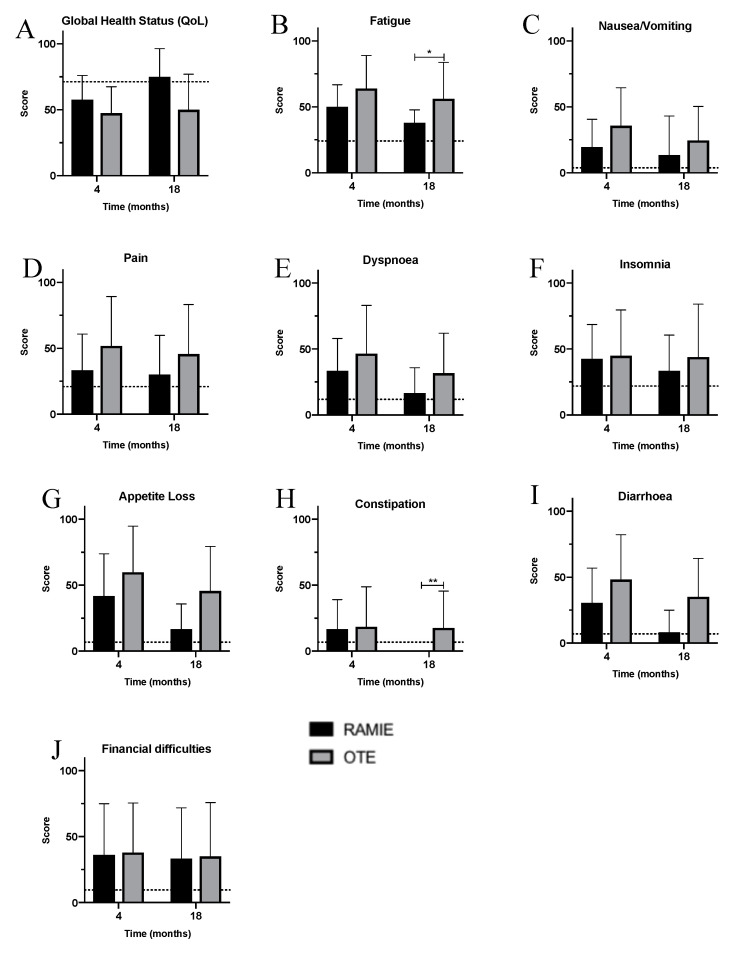
Quality of life and symptoms stratified by cohorts—RAMIE (black) vs. OTE (grey). (**A**) Global health status; (**B**) fatigue; (**C**) nausea/vomiting; (**D**) pain; **E**: dyspnea; (**F**) insomnia; (**G**) appetite loss; (**H**) constipation; (**I**) diarrhea; and (**J**) financial difficulties. OTE: open transthoracic esophagectomy; RAMIE: robot-assisted minimally-invasive esophagectomy.

**Figure 4 jcm-09-03513-f004:**
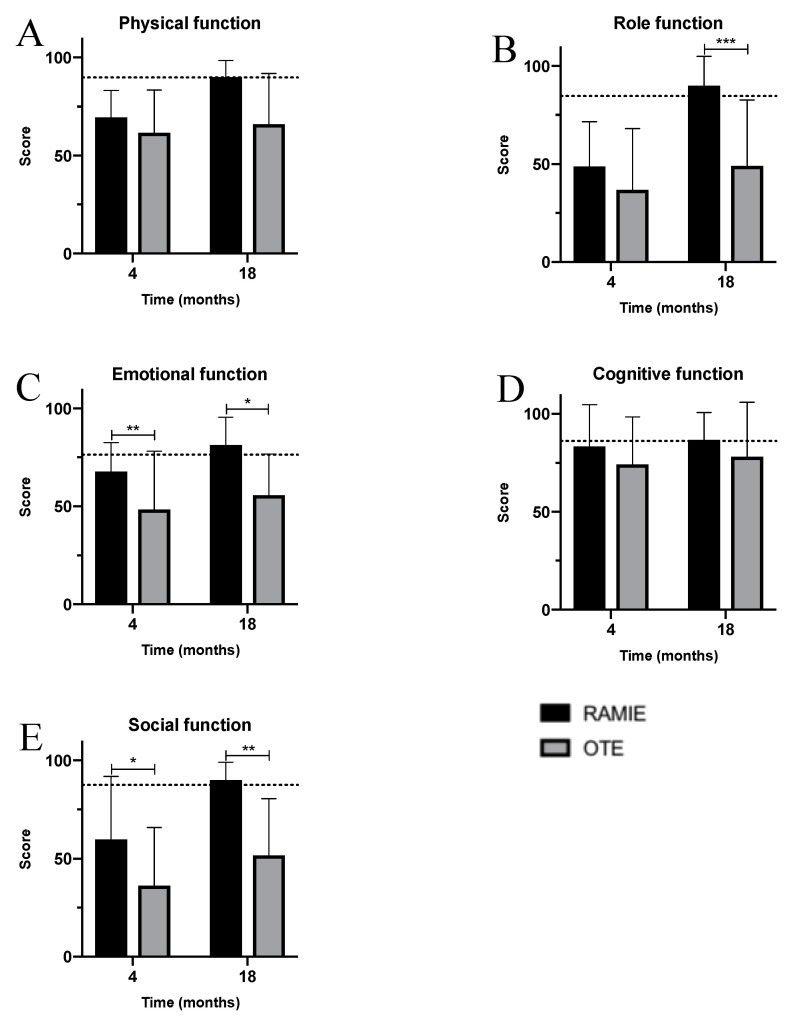
Function stratified by cohorts—RAMIE (black) vs. OTE (grey). (**A**) Physical function; (**B**) role function; (**C**) emotional function; (**D**) cognitive function; and (**E**) social function. OTE: open transthoracic esophagectomy; RAMIE: robot-assisted minimally-invasive esophagectomy.

**Figure 5 jcm-09-03513-f005:**
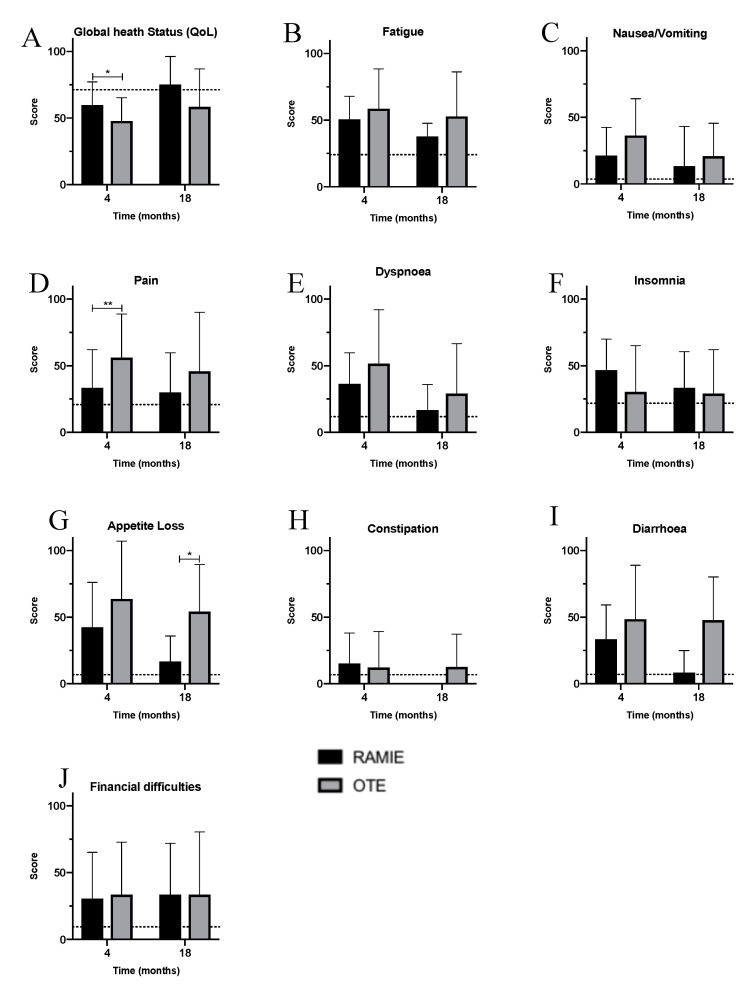
Quality of life and symptoms in the propensity score matching (PSM) analysis stratified by cohorts—RAMIE (black) vs. OTE (grey). (**A**) Global health status; (**B**) fatigue; (**C**) nausea/vomiting; (**D**) pain; (**E**) dyspnea; (**F**) insomnia; (**G**) appetite loss; (**H**) constipation; (**I**) diarrhea; and (**J**) financial difficulties. OTE: open transthoracic esophagectomy; RAMIE: robot-assisted minimally-invasive esophagectomy.

**Figure 6 jcm-09-03513-f006:**
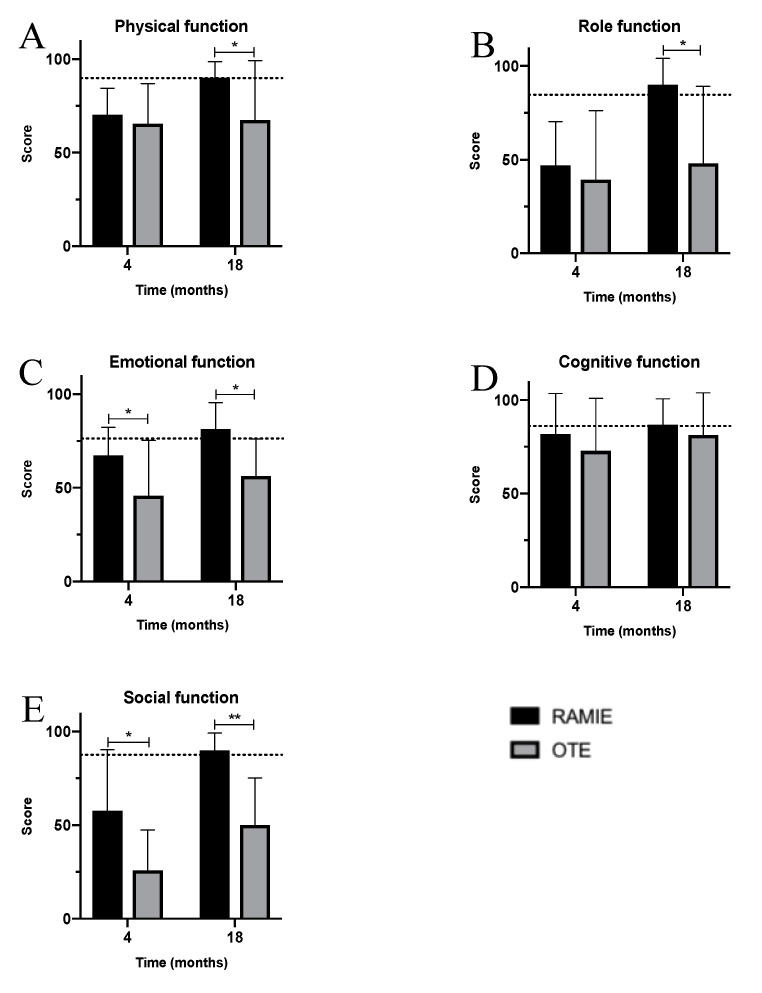
Function in the PSM analysis stratified by cohorts—RAMIE (black) vs. OTE (grey). (**A**) Physical function; (**B**) role function; (**C**) emotional function; (**D**) cognitive function; and (**E**) social function. RAMIE: robot-assisted minimally-invasive esophagectomy; OTE: open transthoracic esophagectomy.

**Table 1 jcm-09-03513-t001:** Baseline comparison of all patient characteristic—cohorts stratified by surgical approach (RAMIE vs. OTE).

	RAMIE (*n* = 12)	OTE (*n* = 29)	*р*-Value
**Age** (years), mean ± SD (range)	64.5 ± 9.1 (50–75)	61.5 ± 8.2 (51–81)	0.332 ^a^
**Gender**, % male	83.3	86.2	0.813 ^b^
**BMI** (kg/m^2^), median ± SD (range)	26.5 ± 4.6 (21.6–34.0)	28.3 ± 4.5 (19.8–44.2)	0.264 ^a^
**Comorbidities**, %			
Arterial hypertension	25.0	55.2	0.078 ^b^
Coronary heart disease	16.7	6.9	0.337 ^b^
Myocardial infarction	0	0	
Heart failure	8.3	6.9	0.872 ^b^
Diabetes	0	17.2	0.965 ^b^
COPD	16.7	10.3	0.574 ^b^
Alcohol consumption	8.3	6.9	0.872 ^b^
Smoking	41.7	24.1	0.262 ^b^
**ASA-classification**, %			0.136 ^b^
I	0	0	
II	25.0	50.0	
III	66.7	50.0	
IV	8.3	0	
**Additional treatment**, %			
Neoadjuvant chemotherapy	75.0	62.1	0.472 ^b^
Neoadjuvant radiotherapy	16.7	10.3	0.574 ^b^
Adjuvant chemotherapy	58.3	40.0	0.295 ^b^
Adjuvant radiotherapy	8.3	7.1	0.896 ^b^

Data presented as mean ± standard deviation (SD), median, min, and max or relative frequencies. Continuous variables were compared using ^a^ Student’s *t*-test, while categorical variables were compared using ^b^ Chi square test. *p*-values of less than 0.05 were considered statistically significant. ASA: American Society of Anesthesiologists; BMI: body mass index; COPD: chronic obstructive pulmonary disease; OTE: open transthoracic esophagectomy; RAMIE: robot-assisted minimally-invasive esophagectomy.

**Table 2 jcm-09-03513-t002:** Surgery-associated characteristics of all patients—cohorts stratified by surgical approach (RAMIE vs. OTE).

	RAMIE (*n* = 12)	OTE (*n* = 29)	*p*-Value
**Total time of surgery** (min), mean ± SD (range)	357.8 ± 86.7 (232–524)	394.5 ± 101.1 (245–589)	0.278 ^a^
**Type of anastomosis**, %			**0.014 ^b^**
End-to-end	8.3	7.1	
End-to-side	33.3	78.6	
Hand-sewn	58.3	14.3	
**Postoperative complications**, %			
0	58.3	62.1	
I	0	27.6	
II	41.7	10.3	
**Histopathological results**, %			0.283 (T) 0.317 (N) 0.125 (M) 0.355 (R) TNM 0.430
pT0	25.0	10.3	
pT1	0	20.7	
pT2	25.0	20.7	
pT3	50.0	48.3	
pN0	50.0	48.3	
pN1	25.0	37.9	
pN2	25.0	6.9	
pN3	0	6.9	
pM0	100.0	82.8	
pM1	0	17.2	
pR0	91.66	93.1	
pR1	8.33	6.9	
**Postoperative tumor stage** (UICC), %			
0	25.0	10.3	
IA	0	17.2	
IB	0	0	
IIA	16.7	6.9	
IIB	0	3.4	
IIIA	25	17.2	
IIIB	33.3	27.6	
IVA	0	3.4	
IVB	0	13.8	
**Size of tumor** (mm), mean ± SD (min-max)	31.9 ± 11.7 (17–46)	20.6 ± 20.9 (2.5–65)	0.164 ^a^
**Lymph nodes harvested, n**	31.0 ± 10.0 (20–46)	18.7 ± 12.1 (7–47)	0.004 ^a^
**Positive lymph nodes harvested, n**	1.4 ± 1.9 (1–5)	1.9 ± 2.7 (1–10)	0.609 ^a^
**Length of stay in hospital** (days), mean ± SD (min-max)	18.9 ± 8.6 (12–42)	15.3 ± 3.5 (9–24)	0.180 ^a^

Data presented as mean ± standard deviation (SD), median, min, and max. Continuous variables were compared using ^a^ Student’s *t*-test (normally distributed), while categorical variables were compared using ^b^ Chi square test. *p*-values of less than 0.05 were considered statistically significant. N: number; OTE: open transthoracic esophagectomy; RAMIE: robot-assisted minimally-invasive esophagectomy; UICC: Union for International Cancer Control.

**Table 3 jcm-09-03513-t003:** Baseline comparison of patient characteristics in the propensity score-matched cohorts stratified by surgical approach (RAMIE vs. OTE).

	RAMIE (*n* = 11)	OTE (*n* = 11)	*p*-Value
**Age** (years), mean ± SD (range)	64.4 ± 9.5 (48–75)	63.2 ± 6.0 (51–73)	0.732 ^a^
**Gender**, % male	81.8	72.7	0.611 ^b^
**BMI** (kg/m^2^), median ± SD (range)	27.0 ± 4.5 (19.8–29.3)	27.8 ± 4.3 (23.1–30.9)	0.651 ^a^
**Comorbidities** (yes), %			
Arterial hypertension	27.3	63.6	0.087 ^b^
Coronary heart disease	18.2	0	0.138 ^b^
Heart failure	0	9.1	0.306 ^b^
Myocardial infarction	0	0	
Diabetes mellitus	18.2	27.3	0.611 ^b^
COPD	18.2	0	0.138 ^b^
Alcohol consumption	9.1	9.1	1.000 ^b^
Smoking	36.4	18.2	0.338 ^b^
**ASA-classification**, %			0.647 ^b^
I	0	0	
II	27.3	36.4	
III	72.7	63.6	
IV	0	0	
**Additional treatment**, %			
Neoadjuvant chemotherapy	72.7	72.7	1.000 ^b^
Neoadjuvant radiotherapy	18.2	9.1	0.534 ^b^
Adjuvant chemotherapy	54.5	25.0	0.198 ^b^
Adjuvant radiotherapy	9.1	0	0.329 ^b^

Data presented as mean ± standard deviation (SD), median, min, and max. Continuous variables were compared using ^a^ Student’s *t*-test, while categorical variables were compared using ^b^ Chi square test. *p*-values of less than 0.05 were considered statistically significant. ASA: American Society of Anesthesiologists; BMI: body mass index; COPD: chronic obstructive pulmonary disease; OTE: open transthoracic esophagectomy; RAMIE: robot-assisted minimally-invasive esophagectomy.

**Table 4 jcm-09-03513-t004:** Surgery-associated characteristics of propensity score-matched cohorts stratified by surgical approach (RAMIE vs. OTE).

	RAMIE (*n* = 11)	OTE (*n* = 11)	*p*-Value
**Total time of surgery** (min), mean ± SD (range)	357.7 ± 91.0 (232–524)	369.4 ± 83.1 (205–460)	0.757 ^a^
**Type of anastomosis, %**			0.190 ^b^
End-to-end	9.1	10.0	
End-to-side	36.4	90.0	
Hand-sewn	54.5	0	
**Type of stapler**, %			
Circular stapler	45.5	100	**0.004** ^b^
Stapler	54.5		
**Postoperative complications**, %			
0	54.5	54.5	
I	0	27.3	
II	45.5	18.3	
**Postoperative tumor stage** (UICC), %			0.881 (T) 0.330 (N) 0 (M) 0.384 (R) TNM 0.952
0	27.3	27.3	
I A	0	0	
I B	0	0	
II A	18.2	18.2	
II B	0	0	
III A	27.3	36.4	
III B	27.3	18.2	
IV A	0	0	
IV B	0	0	
**Size of tumor** (mm), mean ± SD (min-max)	31.8 ± 11.7 (7–48)	16.0 ± 22.24 (3–60)	0.156 ^a^
**Lymph nodes harvested, n**	29.9 ± 9.8 (19–46)	18.1 ± 13.8 (4–40)	0.031 ^a^
**Tumor-positive lymph nodes harvested, n**	1.3 ± 2.0 (1–10)	1.8 ± 3.3 (2–5)	0.640 ^a^
**Length of stay in hospital** (days), mean ± SD (min-max)	19.3 ± 8.9 (12–42)	14.2 ± 2.4 (9–17)	0.082 ^a^

Data presented as mean ± standard deviation (SD), median, min, and max. Continuous variables were compared using ^a^ Student’s *t*-test (normally distributed), while categorical variables were compared using ^b^ Chi square test. *p*-values of less than 0.05 were considered statistically significant. N: number; OTE: open transthoracic esophagectomy; RAMIE: robot-assisted minimally-invasive esophagectomy; UICC: Union for International Cancer Control.

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
