# Peer review of "Long-Term, Health-Related Quality of Life after Open and Robot-Assisted Ivor-Lewis Procedures—A Propensity Score-Matched Study"

_jcm, 2020, doi:10.3390/jcm9113513_

Round 1
Reviewer 1 Report
I read paper.
However, I have very fundamental question. You set gold standard OTE, not MIE, for example, endoscopic surgery by hands.
In my knowledge, we can divide surgeries into two main categories. One is traditional (invasive way), the other is minimally-invasive surgery. And minimally-invasive surgery also can divide into two main categories. By hands, or by robots. By robots one is, in your paper, RAMIE.
If, you compared MIE(by hands) and RAMIE(by robots), this paper has meaning, maybe. But, your paper compared OTE(invasive) and RAMIE(minimally-invasive). I think this comparing is no need to checking. It is already apparent over 10 years ago.
Or if, previous da-Vinci surgery has very big problem, and HRQoL is very bad, and your da-Vinci surgery change it better, it is meaning. But, I cannot find from your introduction.
Author Response
We thank the esteemed reviewer for his comment. We agree on his opinion regarding different types of surgical apporaches and different types of procedures. We all agree that surgery is the best currative treatment option for patients suffering from esophageal cancer [1]. Yet, to our knowledge, no clear recommendation or scientific evidence for one procedure and no ambigouse definition of one surgical procedures exists [2, 3].
OTE was the first currative surgical approach [4]. Since the 1990, novel surgical approaches, often entiteled “minimally invasive“ (MI) were introduced, combining open, laparascopic and recently robotic approaches [2, 5-10]. Yet, if a group for example mentions MIE, this may also include HMIE. The same appeals for RAMIE, including full Robotic, but also hybrid Robotic approaches [6, 7, 10, 11].
Further, the term minimally invasive describes very different techniques:
- So called hybrid apporaches, combining open abdominal and minimally invasive thoracic approaches or hybrid minimally invasive combining a minimally invasive abdominal part and conventional thoracic part [3, 9, 12-14]. Due to the technical limitations of conventional minimally invasive techniques, hybrid approaches are the most commonly used technique.
- Complete minimally invasive (both abdominal and thoracic approach) with different anatomical techniques (Ivor Lewis or McKoewn procedure [6, 11, 15]).
Even high-ranking published studies, which compared open with "minimally invasive" techniques, have used the hybrid technique as minimally invasive technique versus open technique for the thoracic part [8, 9, 14]. The ROBOT trial as well compared the hybrid-robotic approach to the open technique as gold standard from a surgical point of view (ROBOT Trial [7]). Even when comparing HRQoL, HRQoL after OTE serves as reference [16].
The term „minimally invasive“on the other hand is used very inhomogeneously. Further, MIE is a relatively new procedure and known for its long learning curve. We therefore do not consider the minimally invasive technique to be a good reference for a meaningful comparison. Rather, OTE is a well established procedure, we considere it to be adequat for comparison. However, the gold standard for esophageal resection is described very inhomogeneously and more importantly is also lived and published very differently around the world [6, 7, 10, 11]. In many countries esophageal resection is still performed as a completely open procedure and MIE has not been established as new gold standard yet or is about to be established [16-18].
From a psycological point of vie, the use of modern Robot technology could have a frightening effect on the patient: After a successful operation, the patient might develope negative feelings towards anf for having been operated with the help of a machine without the surgeon having operated using his hands and the tissue feeling. Psychological effects, well-being and HRQoL after RAMIE have been neglected so far, mainly reporting on hybrid Robotic procudures [7, 10, 11]. To our knowledge, no comparisson of long term HRQoL after the learning curve of RAMIE and latter exist.
For these reasons, we still consider the relevance of the manuscript to be very high. It especially focusses on the influence of an even more advanced technique (RAMIE) and compares it to the gold standard, thereby maximally focussing on the techniques applied and emphasizing quality of life in the short and long term, depending on the surgical procedure. All other factors - such as perioperative complications, tumor stages stages etc. - have been excluded by the propensity score matching, This allows this explicit and clear comparison of the influence of the surgical procedures on HRQoL, body functions and psychological well-being, also in the longer term. Wilfully, we did not focus on surgical problems and complications, but on HRQoL.
We thank the esteemed reviewer for his comment and have added the following points in the introduction as well as the discussion and the introduction.
Line 57
Old
Postoperative courses and oncological radicalness after RAMIE have been shown to be comparable to OTE [5, 6, 8, 19, 20].
New
Postoperative courses and oncological radicalness after RAMIE have been shown to be comparable to OTE [5, 6, 8, 19, 20]. Yet, no clear definition of the surgical gold standard for EC exists and different techniques are still applied worldwide [1, 2, 18, 21, 22]. Therefore, OTE is often used to as reference procedure, even in large trials like the TIME-, MIRO- and the ROBOT-trial [7-9, 18, 22-24]. Further, terms for esophagectomies, i.e. MIE, HMIE and RAMIE are used heterogenously in the literature, including Ivor-Lewis-, McKewon- and other approaches [2, 10, 13, 25]. Additionally, no surgical approach has unambiguously been proven to be superior compared to the others [3].
Line 327
Old
This analysis further focussed on short- as well as long-term postoperative functions and different body functions after OTE and RAMIE.
New
This analysis further focussed on short- as well as long-term postoperative functions and different body functions after OTE and RAMIE. Since OTE is a well established procedure and the introduction of novel techniques and procedures always has to deal with the learning curve, we only included full OTE and for better comparison full RAMIE in our analysis [2, 9, 18, 25]. Since complications Clavien-Dindo >II are often used as a stopping point, we chose to use it as one of our inclusion criteria [7, 23].
Line 348
Old
To the best of our knowledge, no further study has analysed or compared HRQoL after RAMIE versus OTE.
New
To the best of our knowledge, no further study has analysed or compared HRQoL after RAMIE versus OTE. Since terms, i.e. MIE, HMIE, RAMIE are often mixed and heterogeneous cohorts, including Ivor-Lewis- as well as McKewon, Sweet and other types of surgical approaches are often mixed, we consider our approach, comparing full RAMIE to the established gold standard OTE, only including Ivor-Lewis-procedures, appropriate. Further, most studies report on hybrid procedures and avoid the full laparoscopic approach due to the limitations of laparoscopic surgery [14]. Our RAMIE-cohort on the other hand underwent a full robotic approach after having completed the learning curve in hybrid RAMIE [26].
Line 397
Old
Interestingly, they even demonstrated an “over recovery” of emotional and social function after surgery, with both functions being reported as superior to preoperative levels [27].
New
Interestingly, they even demonstrated an “over recovery” of emotional and social function after surgery, with both functions being reported as superior to preoperative levels [27]. Klevebro et al. on the other hand reported no differences in HRQoL when comparing OTE to MIE and HMI up to one year after surgery [16].
English language and style
(x) Extensive editing of English language and style required
We are a bit surprised about the hint that extensive English improvement is required. As described in the manuscript, the manuscript was corrected and edited by a native speaker (see note Acknowledgements). We hope the changes made now meet your requests.
Line 442
Old
The sponsors had no role in the design, execution, interpretation, or writing of the study.
New
The sponsors had no role in the design, execution, interpretation, or writing of the study. We acknowledge the help of Deborah Nock, Medical Writeaway, for English corrections.

Reviewer 2 Report
I want to congratulate the authors on their manuscript. They compared HRQOL outcomes after esophagectomy (robotic vs. open) and demonstrated an improvement in HRQOL over time. In the authors' attempt to remove bias from their study, patients with major complications were excluded.
My comments are:
1. Do major complications contribute to HRQOL more than minor complications?
2. Can you remove the bias of complications on HRQOL without introducing bias by patient selection and therefore unable to recommend one treatment over another?
3. Does stratifying patients by minor complications vs. major complications address the above questions and strengthen the author's conclusions?
Author Response
- Do major complications contribute to HRQOL more than minor complications?
We thank reviewer 2 for this very important and interessting question. Derorgar et al. previously identified a higher and clinically relevant influene of major complications on HRQoL in patients undergoing OTE [28]. It has further been shown that major complications after OTE highly influence HRQoL negatively up to ten years after surgery [28, 29]. Scarpa et al. identified reduced emotional and physical function, both in the short and long term, in patients experiencing complications after OTE [30]. Yet, Scarpa et al. did not specifically differentiate between minor and major complications, respectively on HRQoL. We, on the other hand, have previously shown, that patients undergoing esophagectomy and suffering from postoperative complications Clavien-Dindo >II, report significantly better HRQoL compared to patients without (major) postoperative complications. This so-called “response shift“ seems to highly influence the perception of the postoperative course and postoperative courses [31]. Further, patients adapt to new situations, i.e. physical impairments, and reconceptualize [29]. Yet, we have to acknowledge that the studies mentioned above all analysed clinical courses after OTE, which is a highly invasive open surgical approach causing a large operative trauma.
Most studies report on mayor complications or do not differentiate between minor and mayor complications [28-30]. By our inclusion criteria we aimed to exclude this vagueness of only using the term „complications“and the response shift influencing the results. However, in order to keep cohort most homogenouse and not overestimate the influence of major complications and prolonged postoperative stays due to complications, we did not include the latter in our analysis. Our inclusion criteria voluntarily only included patients suffering from minor complications as we believe, in accordance with others, that major complications influence HRQoL differently, not eventually more [14]. Both the MIRO- and the ROBOT-trial used complications Clavien-Dindo
- Can you remove the bias of complications on HRQOL without introducing bias by patient selection and therefore unable to recommend one treatment over another?
We thank the esteemed reviewer and acknowledge his concerne. Due to the study design, extracting data from a prospectively maintained database, selection bias is primarily minimized during the inclusion period. Nonethelesse, we agree with reviewer 2 that a certain bias is introduced by our selection. However, it is assumed that minor complications tend to be underreported as patients might suffer less and perhaps do not really notice the complication, i.e. urinary tract infections or a little coughing [14]. Comparing a cohort with potentially underrepresented minor complications to a cohort with relatively overrepresented major complications would not represent reality. In order to only identify the influence of the procedure on HRQoL including one type of complication, we considere our selection approriate.
Patients suffering from mayor complications have to stay in hospital for a longer period of time, achieve additional treatment, therapy and care, consequently really notice the difference and potentially take advantage of this situation or even benefit from it, so called secondary morbid gain [30, 31]. Kaupilla et al. identified a higher deterioritation in patients suffering from medical complications up to ten years after surgery and up to five years for patients suffering from postoperative surgical complications after OTE [32]. Medical and surgical complications are not fully comparable to minor and major complications, but may be used as surrogates in this context. Yet, Kaupilla et al. did not perform a comparisson between the two cohorts. Stratifiying patients by this approache as a new point of view underlines our approach of comparing patients suffering from minor complications. By comparing two different cohorts (minor vs. major or medical vs surgical complications, respecitvly) with regards to HRQoL one would rather introduce a bias instead of avoiding the bias as the therapeutic consequences of either type of complication are very different and might influence HRQoL as well. Further, our main aim writing this paper was the comparission of HRQoL after RAMIE and OTE. When including both types of complications, the comparisson between procedures might be biased. Kaupilla et al. and Derogar et al. used one surgical approach when comparing different aspects [28, 29, 32].
- Does stratifying patients by minor complications vs. major complications address the above questions and strengthen the author's conclusions?
We thank the esteemed reviewer for this interessting question. We think that primarily stratifying cohorts by complications allows for a better comparisson since cohorts were definded more clearly and the central question of this paper if long-term HRQoL differs between the two surgical approaches may be answered more clearly without having to taking potential confounders, such as type of complication, into consideration. However, we agree with reviewer 2 that it would be more than interessting to compare patients suffering from major complications in both cohorts as we compared patients suffering from minor complications in this paper. The additional longterm comparisson, independantly of the type of complication, would be of interessted as well.
Line 408
Old
This may also be attributable to the improved postoperative HRQoL and self-perception.
New:
This may also be attributable to the improved postoperative HRQoL and self-perception. Additionally, not only surgery and the surgical approach chosen influence postoperative HRQoL, but complications also influence the latter. Heits et al. reported on the so-called “response shift” after prolonged postoperative courses due to mayor complications, while Scarpa et al. mentioned impaired postoperative HRQoL on the long term in patients suffering from postoperative complications [30, 31]. Most studies only report on mayor complications or do not differentiate between minor and mayor complications [28-30]. Rutegard et al. as well reported on impaired HRQoL after mayor surgical complications in patients undergoing OTE [33]. Kaupilla et al. on the other hand reported on an influence of medical complications up to 10 years after surgery and surgical complications up to 5 years after surgery [32]. By our inclusion criteria we aimed to exclude this vagueness and the response shift influencing the results. Our inclusion criteria voluntarily only included patients suffering from minor complications in order to avoid response shift, an extensive influence of mayor complications or following extensive or long therapy, i.e. esosponge-therapy, stent, reoperation, on HRQoL [14, 28, 29, 32]. However, in this context, it would be especially interesting to see if mayor complications influence HRQoL in RAMIE patients compared to OTE patients differently, especially in the long term.
